# Microscopic and Biochemical Hallmarks of *BICD2*-Associated Muscle Pathology toward the Evaluation of Novel Variants

**DOI:** 10.3390/ijms24076808

**Published:** 2023-04-06

**Authors:** Andreas Unger, Andreas Roos, Andrea Gangfuß, Andreas Hentschel, Dieter Gläser, Karsten Krause, Kristina Doering, Ulrike Schara-Schmidt, Sabine Hoffjan, Matthias Vorgerd, Anne-Katrin Güttsches

**Affiliations:** 1Department of Cardiovascular Medicine, Institute for Genetics of Heart Disease (IfGH), University Hospital Münster, 48149 Münster, Germany; 2Institute of Physiology II, University of Münster, 48149 Münster, Germany; 3Department of Neurology, Heimer Institute for Muscle Research, University Hospital Bergmannsheil, Ruhr University Bochum, 44789 Bochum, Germany; 4Department of Pediatric Neurology, Centre for Neuromuscular Disorders, Centre for Translational Neuro- and Behavioral Sciences, University Duisburg-Essen, 45122 Essen, Germany; 5Children’s Hospital of Eastern Ontario Research Institute, Ottawa, ON K1H 8L1, Canada; 6Leibniz-Institut für Analytische Wissenschaften-ISAS-e.V., 44139 Dortmund, Germany; 7Genetikum, Center for Human Genetics, 89231 Neu-Ulm, Germany; 8Department of Human Genetics, Ruhr-University Bochum, 44801 Bochum, Germany

**Keywords:** *BICD2*, muscle proteomics, endoplasmic/sarcoplasmic reticulum, Golgi pathology, biglycan, thrombospondin-4

## Abstract

*BICD2* variants have been linked to neurodegenerative disorders like spinal muscular atrophy with lower extremity predominance (SMALED2) or hereditary spastic paraplegia (HSP). Recently, mutations in *BICD2* were implicated in myopathies. Here, we present one patient with a known and six patients with novel *BICD2* missense variants, further characterizing the molecular landscape of this heterogenous neurological disorder. A total of seven patients were genotyped and phenotyped. Skeletal muscle biopsies were analyzed by histology, electron microscopy, and protein profiling to define pathological hallmarks and pathogenicity markers with consecutive validation using fluorescence microscopy. Clinical and MRI-features revealed a typical pattern of distal paresis of the lower extremities as characteristic features of a *BICD2*-associated disorder. Histological evaluation showed myopathic features of varying severity including fiber size variation, lipofibromatosis, and fiber splittings. Proteomic analysis with subsequent fluorescence analysis revealed an altered abundance and localization of thrombospondin-4 and biglycan. Our combined clinical, histopathological, and proteomic approaches provide new insights into the pathophysiology of *BICD2*-associated disorders, confirming a primary muscle cell vulnerability. In this context, biglycan and thrombospondin-4 have been identified, may serve as tissue pathogenicity markers, and might be linked to perturbed protein secretion based on an impaired vesicular transportation.

## 1. Introduction

BICD cargo adapter 2 (*BICD2*) is a ubiquitously expressed protein that initiates motor protein-driven directional cargo movement and is thus crucial for proper cellular protein transport [1]. It is an evolutionary conserved motor-adaptor protein that consists of three highly conserved coiled-coil regions: CC1, CC2, and CC3 [2]. The N-terminal domain interacts with the dynein motor, whereas the C-terminal domain binds to various cargoes, such as RAB6A [2]. Interaction of mutant BICD2 with the dynein-dynactin complex and/or with RAB6A leads to fragmentation of the Golgi apparatus, resulting in perturbations of BICD2-dynein-dynactin-mediated vesicular trafficking that may have an impact on regular development and maintenance of motor neurons [3,4,5,6,7].

Along this line, pathogenic variants within the *BICD2* gene have been described as the cause of spinal muscular atrophy with lower extremity predominance 2 (SMALED2) and hereditary spastic paraplegia (HSP) [2,7,8,9,10,11]. Recently, we described two independent German families presenting with a myopathy predominantly affecting lower extremities but also defined by mild scapula alata and paresis of trunk muscles. These clinical findings were associated with two different pathogenic variants in the *BICD2* gene [12]. Histopathological evaluation of the skeletal muscle biopsies revealed dystrophic changes accompanied by a high abundance of secretory vesicles between fragmented Golgi and enlarged Sarcoplasmic/Endoplasmic Reticulum (SR/ER) [12]. Those findings, which hint toward a primarily myopathic genesis of the disease, have been confirmed by phenotypical and molecular characterization of a mouse model with *Bicd2* knockout in the muscle [13], indicating that a loss of BICD2 from muscle tissue is involved in the motor neuron pathology in SMALED2 patients [13].

Next generation sequencing (NGS) offers the molecular analysis of a variety of genetic disorders in parallel. However, this approach frequently results in the identification of variants of unknown significance, for which the evaluation of their pathogenicity and thus of their impact on the clinical manifestation is challenging. Although in silico analyses represent valuable tools to classify the pathogenicity and their possible effects on protein function, they lack the possibility to determine histopathologic alterations, effects on protein interactions, and dysfunction. An examination of the pathophysiological effects thus often requires additional experiments on vulnerable cellular populations such as muscle cells. By taking known cellular functions of BICD2 (including the recently published data indicating a primarily myogenic pathology [13]) into consideration, we therefore investigated skeletal muscle biopsies derived from *BICD2* patients, including one with an already known pathogenic variant and six with novel *BICD2* missense variants using histology, immunofluorescence analysis, electron microscopy, and protein profiling to define pathogenicity markers (Figure 1). By doing so, we intended to gain insights into whether the novel *BICD2* missense variants might be pathogenic or rather represent benign amino acid substitutions.

## 2. Results

### 2.1. Clinical and MRI Features of Patients with BICD2-Associated Myopathy Reveal Typical Pattern of Muscle Weakness of the Legs (with Sparing of Adductor Muscles)

The most prominent clinical feature of *BICD2*-associated myopathy is an atrophic paresis of the leg muscles, which was pronounced in the distal leg muscles of patients p1, p3, p4, p5, and p6 and in the proximal and distal leg muscles of p2 (Table 1). In contrast, p7 had a proximally pronounced paresis of the legs (Table 1). The labeling of the patients presented in this study is used according to Table 1 throughout the manuscript.

MRI of the leg muscles was performed on p1, p5, and p7; a data report is available on p2. Figure 2 shows severe fatty degeneration of the quadriceps muscles, semimembranosus, biceps femoris, and dorsal lower leg muscle changes in p1. To compare, the lower extremities of p5 had symmetrical, distally pronounced fatty involution of the quadriceps femoris, biceps femoris, caput longum, and thigh muscles as well as symmetrical fatty degeneration of the gastrocnemius, soleus, peroneus, and tibialis anterior in the calf muscles (Figure 2). There was no edema in the leg muscles. The adductor muscles and extensor digitorum longus muscles were spared from fatty degeneration (Figure 2). MRI data from p2 (not performed in our hospital) showed similar results with massive fatty degeneration of all leg muscles, sparing the adductor muscles.

MRI of the lower extremities of p7, who also carries a variant in the *COL6A1* gene in addition to the variant in the *BICD2* gene, displayed slightly asymmetric, distally pronounced fatty degeneration of the quadriceps femoris (arrows), biceps femoris, gastrocnemius, and left peroneus longus muscles (Figure 2). To compare, a patient with a genetically confirmed *COL6A1*-associated myopathy was included: note the pronounced degeneration and atrophy of the distal vastus lateralis and medial gastrocnemius muscles, which is similarly found in p7 (Figure 2, arrows).

### 2.2. Genetic Analysis Revealed a Known Pathogenic and Four Novel Sequence Variants in BICD2

Genetic analysis of p1 revealed the heterozygous pathogenic missense variant c.320C>T p.(Ser107Leu), which has already been described in the literature [12]. Additionally, four novel *BICD2* missense variants were identified (Table 1). These variants are (i) collectively rare in population databases (except for the Lys818Glu variant, which was reported eight times in GnomAD); (ii) not described in the literature; and (iii) classified as variants of unknown significance (for detailed information, see Appendix A). Genetic analysis of p2 revealed the heterozygous rare missense variant c.1195C>T p.(Arg399Cys). In p3, a whole exome sequencing analysis led to the identification of the heterozygous missense variant c.2189G>A, p.(Arg730His), which was confirmed also in p4, his clinically affected father (Table 1). Genetic analysis of p5 and her daughter, p6, revealed the heterozygous sequence variant c.1904G>T p.(Arg635Leu) in *BICD2* and additionally the heterozygous sequence variant c.2272G>A p.(Val758Met) in *FLNC*, which was also classified as variant of uncertain significance. In the unaffected daughter (II.2 of family 4), neither sequence variant was detected (Appendix A).

Genetic analysis of p7 (Table 1) revealed the heterozygous sequence variant c.2452A>G, p.(Lys818Glu) in *BICD2* in addition to the heterozygous sequence variant c.1694G>A, p.(Arg565Gln) in *COL6A1*. Moreover, the variants c.280C>T, p.(Arg94Trp) in *PYGM* and c.1084A>G, p.(Lys362Glu) in *ACADM* were detected. However, both variants are mono-allelic, and the respective genes are known to cause recessive disorders. The patient’s parents were already deceased and therefore unavailable for genetic testing. According to the patient, they did not suffer from neuromuscular problems. The *COL6A1* variant was also classified as of uncertain significance, with the affected nucleotide position not being highly conserved among species. The unaffected siblings (II.2 and II.3, Appendix A) neither carried the *BICD2* nor the *COL6A1* variant.

### 2.3. Histopathologic Analysis Reveals Slight Myopathic Changes in Novel BICD2-Patients

Muscle pathology in p1, p2, p3, p5, and p7 revealed myopathic features, which were most prominent in p1. Histopathological parameters, including the detailed results of light microscopic analysis as well as immunofluorescence (IF) and Western blot (WB) results, are listed in Table 2. Representative images of hematoxylin-eosine (H&E), Gömöri trichrome (TC), and ATPase staining showing the muscle pathology findings in p1, p3, p5, and p7 are shown in Figure 3.

### 2.4. Electron Microscopy (EM) Showed Myopathic Changes including Abundant Autophagic Vacuoles

In addition to light microscopic analysis, electron microscopy of skeletal muscle tissue from p1, p3, p5, and p7 was performed and revealed advanced stages of myofibrillar breakdowns, lesions, and a massive presence of lysosomal vesicles, accompanied by enlarged polymorphic mitochondria and abundant autophagic vacuoles (Figure 4). In all patients, especially in p3, the contractile apparatus showed massive perturbations of structural integrity.

### 2.5. Proteomic Analysis Revealed an Increase of Proteins Associated with Perturbed Vesicular Transport

Next, we aimed to identify novel protein markers that were (i) indicative of the pathogenicity of *BICD2* variants and (ii) in accordance with known functions of the corresponding protein in terms of a pathophysiological connection. To this end, we filtered proteomic data obtained from whole protein extracts of the muscle biopsy of p1 for functional candidates, including substrates of perturbed versicular transport. In this context, the proteomic data revealed a statistically significant increase of thrombospondin-4 and biglycan (Figure 5), two extracellular matrix proteins secreted by the vesicular transport machinery.

### 2.6. Verification of Proteomic Data by Immunofluorescence Analyses

The two candidate markers unraveled by our proteomic discovery approach, thrombospondin-4 (THBS4) and biglycan, were next studied using immunofluorescence analyses (Figure 6). Prompted by the known pathology of the Golgi system in *BICD2*-pathophysiology and the fact that the Golgi represents a functional continuum with the SR/ER and the intermediate compartment (ERGIC-3), we additionally analyzed SEC63, BET1, ERGIC-3, and Golgin-97 in p1 (with described pathogenic variant c.320C>T) in addition to p3, p5, and p7, who were carrying novel heterozygous variants (of unknown significance) in *BICD2*. This approach aimed to further elucidate typical *BICD2*-associated alterations of pathophysiological impact. Age-matched controls of patients with neurogenic muscle alterations and normal disease controls were included, respectively (Figure 6). Patients p1, p3, and p5 showed increased sarcoplasmic immunoreactivity of thrombospondin-4. Moreover, biglycan was increased in the peri- and endomysium of p1, p3, and p5. Aggregates with increased immunoreactivity of SEC63 were present (although to different extents) within the sarcoplasm of muscle fibers in the biopsy specimens derived from p1, p5, and p7 (Figure 6). BET1 was slightly increased in the sarcoplasm of muscles derived from p1, p3, and p5, whereas ERGIC-3 and Golgin-97 were enhanced in the endomysial cells of p1 (Figure 6). Of note, SEC63 immunoreactive aggregates were also identified in muscle cells derived from (age-matched) patients presenting with neurogenic muscular atrophies (serving as disease controls) (Figure 6).

## 3. Discussion

Here, we present a comprehensive characterization of the effects of *BICD2*-deficiency in human skeletal muscle tissue, including clinical, MRI, and microscopic features. The results of later studies revealed organelle pathology like Golgi fragmentation accompanied by altered abundance and localization of ER/Golgi-marker proteins. These myopathological findings associated with *BICD2* pathophysiology go beyond an exclusive neuronal degeneration and thus support the findings obtained on a mouse model with a muscle-specific knockout of *Bicd2* [13]. Based on the results of our combined proteomic and histopathological approaches, we provided new insights into the pathophysiology of *BICD2* in skeletal muscle tissue.

On a proteomic level, we identified an increase of proteins that might represent targets of vesicular disturbance: thrombospondin-4 and biglycan are both glycoproteins synthetized in the ER/SR-Golgi network and secreted to the extracellular space by vesicle transport and mediating cell-to-cell and cell-to-matrix interactions. Of note, thrombospondin-4 also plays a role in ER stress response via interaction with ATF6, a major transducer of the unfolded protein response [14]. Given that our microscopic studies revealed altered ER structures and the presence of sarcoplasmic aggregates immunoreactive for SEC63 (an ER-membrane resident protein mediating co- and post-translational transport of nascent polypeptides to the ER), one might speculate that the altered abundance and distribution of thrombospondin-4 might also arise from perturbed ER homeostasis. Indeed, BET1 (SNARE protein involved in the docking process of ER-derived vesicles with the cis-Golgi membrane) was moreover increased in the muscle tissue of patients with *BICD2*-associated myopathy (p1, p3, and p5). This indicates a perturbed vesicle transport from the ER to the Golgi apparatus and thus molecularly links the Golgi and vesicle pathology already known to be crucial in SMALED to ER/SR vulnerability thus expanding the spectrum of subcellular organelles affected by the disease. However, as the SEC63 increase observed in some of our *BICD2* patients also appears in patients with neuromuscular atrophy, this pathomorphological finding might rather be regarded as non-specific and more related to muscle cell denervation than to primary myopathology.

In the past, pathogenic variants in *BICD2* were first described as being causative for SMALED2 or HSP [2,7,8,9,13]. These neurological disorders differ from each other regarding their clinical presentation: patients with SMALED2 present with muscle weakness and muscular atrophy as a sign of lower motor-neuron predominance, whereas patients with HSP show a spastic paresis indicating upper motor-neuron involvement [9,13]. In this context, it is important to note that the location of the underlying pathogenic variant in *BICD2* has been shown to affect the clinical presentation [9]. The pathophysiology of SMALED2 could be linked to mutations in either the N-terminal CC1- or the C-terminal CC3-domain, both of which lead to a disruption of BICD2 binding to RAB6A, a regulator of Golgi–ER trafficking. The disrupted molecular interplay between BICD2 and RAB6A is postulated to result in a disturbed vesicular transport of cargo proteins to the plasma membrane [2,15]. Different from *BICD2* variants associated with SMALED2, manifestation of an HSP phenotype rather seems to be associated with mutations in CC2, the second coiled-coil domain of BICD2 [13]. This region is known to interact with kinesin 1 [16,17,18]. As mutations of the kinesin-1-encoding gene *KIF5A* were linked to manifestation of familial hereditary spastic paraplegia, it can be assumed that altered interactions of BICD2 with kinesin 1 are resulting in a different pathophysiology that is associated with HSP [18]. 

Recently, a murine model of *BICD2*-associated myopathy has been published [13]. By phenotyping this murine model, Rossor and colleagues demonstrated that BICD2 is required for the physiological flow of constitutive secretory cargoes from the trans Golgi network to the plasma membrane. Loss of BICD2 leads to a decrease of vesicle transport from the Golgi to the plasma membrane [13]. Our immunofluorescence findings focusing on proteins resident to the SR/ER-Golgi system support our electron microscopic observations. Altered architecture of the SR/ER-Golgi network may affect its proper function in protein production, including the vesicle-mediated protein transportation to subcellular destinations such as the cleft of the neuromuscular junction. Thus, one might assume that the described decrease of neurotropin release and the consecutive axon degeneration described by Rossor and colleagues secondarily result from perturbed SR/ER-Golgi integrity as a primary myopathological event in SMALED. Of note, thrombospondin-4 is mainly restricted to skeletal muscle and cardiac tissue and is induced with injury or disease [19,20]. Moreover, it accumulates at the neuromuscular junction (NMJ) and at certain synapse-rich structures in adult mice [19]. Of note, biglycan is an extracellular muscle-specific receptor tyrosin kinase (MuSK) binding protein important for NMJ stability [21]. Taken together, our data indicated a similar mechanism also in humans and unraveled two candidate proteins, thrombospondin-4 and biglycan, that might bridge the muscle to the nerve/neuron pathology by hinting toward an altered neuromuscular transmission arising from ER/SR-Golgi pathology. However, further functional studies, including modulation of this pathology with a focus on restoration of proper thrombospondin-4 and biglycan distribution, are needed to prove this hypothesis. This aspect may also be of therapeutic relevance as several FDA- and EMA-approved drugs known to address ER-Golgi function exist. Within this study, we moreover aimed to evaluate if the abovementioned proteins are suitable to serve as pathogenicity markers in patients with missense variants in *BICD2* of unknown significance. Immunofluorescence analyses in p3 and p5 revealed increased immunoreactivity of thrombospondin-4, biglycan, SEC63 (p5), and BET1, comparable to the patient with the known pathogenic *BICD2* mutation (p1). Moreover, the clinical and especially histopathological results in p3 and p5 as well as the MRI data (of p5) are comparable to those of p1. The *FLNC* variant in p5 has only once been described as a variant of unknown significance (see also Appendix A) [22,23]. As the clinical and radiological, as well as histopathological and electron microscopic, data hint at a *BICD2*-associated pathology, we postulate that the *BICD2* missense variant in p5 is rather to be regarded as causative. In contrast, p7 showed alterations of neither thrombospondin-4 nor of biglycan. Here, the results of the immunofluorescence analysis of SR/ER- and Golgi-proteins (Figure 6), in accordance with clinical, histopathological, and MRI results, indicated that the *BICD2* variant is rather to be evaluated as benign. In line with this, this variant was recently classified as likely benign in ClinVar. Additionally, the MRI results indicated that the *COL6A1* variant might explain the manifestation of a neuromuscular disease in this patient. Thus, the two proteins, biglycan and thrombospondin-4, affected by pathogenic *BICD2* mutations, might hold the potential to serve as marker proteins (of profound pathophysiological relevance), enabling evaluation of the pathogenicity of *BICD2* variants, an important aspect in diagnosis and genetic counselling.

## 4. Materials and Methods

### 4.1. Patients and Muscle Biopsies

Seven affected patients from five independent German families with muscle weakness underwent clinical examinations with review of medical notes and previous investigations. The pedigrees of the families are shown in Appendix A. The clinical features for all patients are summarized in Table 1. Briefly, the patients ranged in age from 13 to 70 years at the date of clinical examination. Disease onset was either congenital or in early childhood or adulthood and manifested as slowly progressive muscle weakness. Distal muscle weakness without sensory symptoms was the most prominent clinical symptom in all patients except for p7, who presented with proximal muscle weakness. Creatine kinase (CK) values were slightly elevated, ranging from 150 to 650 U/L in six of the seven patients (Table 1). All patients maintained normal cardiac and respiratory functions. 

The study was approved by the local Ethics Committees of the Ruhr University Bochum (reg. no: 5118-14) and the University Hospital Essen (reg. no: 19-9011-BO).

### 4.2. Muscle MRI

Patients p1, p5, and p7 underwent MRI imaging of the lower extremity muscles in a 1.5-T scanner with a four-channel phased-array coil (Magnetom Symphony, Siemens, Munich, Germany) using a standardized protocol (T1-weighted spin-echo (TR/TE 500/20 ms, slice thickness 10 mm, and matrix 512 × 512) and two T2-weighted STIR (TR/TE 4020/68 ms and 3040/27 ms, respectively; TI 150 ms; slice thickness 10 mm; and matrix 512 × 512). Patient p2 underwent MRI in another hospital. Thus, although a respective report of MRI findings was available, figures were not available and could consequently not be included in Figure 2.

### 4.3. Genetic Analyses

In p2, p3, p5, and p7, a diagnostic gene panel, or whole exome sequencing, was conducted based on next-generation sequencing (NGS) technology using an Illumina system that included the following number of genes known to be causative for the manifestation of neuromuscular disorders: 252 in p2, the whole exome (whole exome sequencing, WES) in p3, 154 in p5, and 235 in p7. Sequence analysis was performed with the software SeqNext (JSI medical systems GmbH, Ettenheim, Germany) and GSVar using the reference genome “UCSC Genome Browser, hg19, GRCh37”. Genetic variants with an allele frequency of more than one percent were discharged, as long as they were not annotated as pathogenic or likely pathogenic in HGMD or ClinVar. Potential pathogenic variants were confirmed via Sanger sequencing, and segregation analyses were subsequently performed in family 1, as described in [12]; family 3, II.1 (p6); family 4, II.2; and family 5, II.2 and II.3 (see Appendix A). Additional information regarding the genetic analysis including the prediction programs used is given in Appendix A.

### 4.4. Histological Studies and Immunofluorescence Stainings

For histological analysis, skeletal muscle biopsies were divided into 0.5 cm³ pieces. For light microscopy, pieces were embedded into Tissue Freezing Medium (Leica Microsystems, Wetzlar, Germany) and snap frozen in liquid nitrogen-cooled isopentane. Hematoxylin-eosin (H&E), Gömöri trichrome (TC), and ATPase staining (ATPase 4.3 and ATPase 9.6) were performed according to standard procedures [24]. 

Immunostaining on muscle biopsies was performed as described previously [25]. The following primary antibodies were used: anti-thrombospondin-4 (1:500, No AB176116, Abcam, Cambridge, UK); anti-biglycan (1:100, No PA5-76821, Thermo Fisher, Darmstadt, Germany); anti-BET1 (1:150, PA5-88961, Thermo Fisher, Darmstadt, Germany); anti-SEC63 (1:100, No NBP1-59694, Novus Biologicals, Centennial, CO, USA); anti-ERGIC-3 (1:100, No GTX122511, GeneTex, San Antonio, TX, USA); and anti-Golgin-97 (1:100, No GTX14445, GeneTex, San Antonio, TX, USA). Secondary antibodies were Cy3- or FITC-conjugated IgG (1:400; Rockland) and incubated o/N 4 °C, including the nucleus marker DAPI (4′,6-diamidino-2-phenylindole, Sigma Aldrich, Darmstadt, Germany).

### 4.5. Electron Microscopy (EM)

Fixed muscle samples from p1, p3, p5, and p7 were cut into longitudinal sections with a vibratome (VT 1000S, Leica, Germany), rinsed twice in PBS, treated with OsO_4_, and counterstained with uranyl acetate in 70% ETOH following prior dehydration and embedded in Durcupan resin (Fluka, Sigma-Aldrich, Taufkirchen, Germany). Resin blocks were polymerized, and ultrathin sections were prepared with a Leica Ultracut S. Sections were adsorbed to glow-discharged formvar-carbon-coated copper grids. Ultrastructural analysis was performed with a Zeiss LEO 910 electron microscope, and images were taken with a TRS sharpeye CCD Camera (Troendle).

### 4.6. Proteomic Analysis

Proteomic profiling on quadriceps muscle derived from p1 and two healthy (sex- and age-matched) controls was carried out on whole muscle protein extracts in a data-independent-acquisition mode, as described previously [25]. In this context, it is important to note that this patient carried the known pathogenic *BICD2* mutation c.320C>T (p.Ser107Leu), which has been acknowledged as a causative mutation for *BICD2*-associted myopathy previously [12] and is thus suitable to identify protein dysregulations of pathophysiological relevance for *BICD2*-associated muscle pathology. Relative protein/peptide abundances were measured in technical triplicates each, and mean values with standard deviations were calculated using Microsoft Excel.

## 5. Conclusions

In this study, we extended the spectrum of potentially pathogenic genetic variants of *BICD2* and provided further evidence that the pathogenicity of *BICD2* mutations is not restricted to neurogenic changes but plays a role in muscle pathophysiology. In this context, biglycan and thrombospondin-4 might represent a molecular bridge of muscle and nerve/neuron vulnerability and hold the potential to serve as tissue pathogenicity markers.

## Figures and Tables

**Figure 1 ijms-24-06808-f001:**
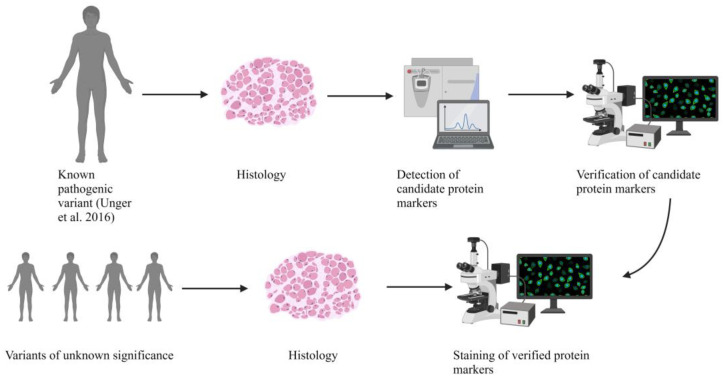
Graphical abstract showing the workflow of the presented study [12]. Created with BioRender.com.

**Figure 2 ijms-24-06808-f002:**
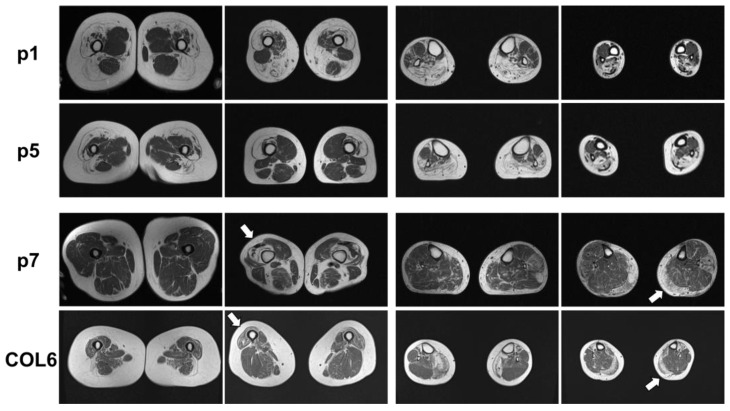
Muscle MRI of p1, p5, and p7 as well as a patient with a pathogenic *COL6A1* mutation. p1: Muscle MRI of p1 with known *BICD2*-associated myopathy showed fatty degeneration of the thigh muscles, pronounced in the lateral biceps femoris muscles (left panels) and the calf muscles, pronounced in the soleus and gastrocnemius muscles (right panels). p5: Muscle MRI of p5 with novel *BICD2* and *FLNC* variants showed fatty degeneration of the vastus lateralis and medialis muscles as well as the lateral biceps femoris muscle in the thigh (left panels) and fatty degeneration of the gastrocnemius and soleus muscles (right panels). p7: Muscle MRI of p7 with novel *BICD2* and *COL6A1* variants showed distal fatty degeneration of the thigh muscles (especially vastus lateralis (left panels, white arrow)) and asymmetric, left, and distally pronounced fatty degeneration in the gastrocnemius muscle, caput mediale, soleus, and peroneus longus muscles (right panels, white arrow). COL6: Muscle MRI of a patient with *COL6A1*-associated myopathy shows typical distal atrophy of vastus lateralis muscle (left panels, white arrow) and gastrocnemius muscles (right panels, white arrow). White arrows indicate muscle degeneration typically associated with *COL6A1*-associated myopathy.

**Figure 3 ijms-24-06808-f003:**
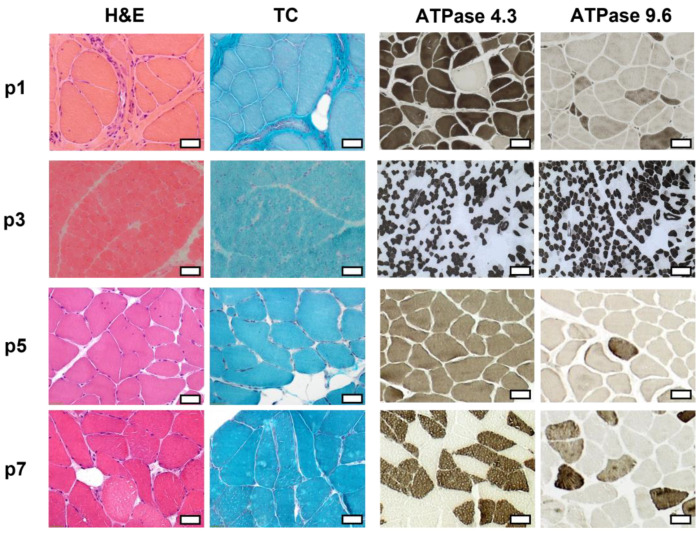
Histopathologic analysis of the skeletal muscle biopsy of p1, p3, p5, and p7. Hematoxylin-eosine (H&E), Gömöri trichrome (TC), and ATPase stainings 4.3 and 9.6 are displayed. Light microscopic analysis revealed primarily myopathic features, which were most prominent in p1. Scale bar: 50 µm.

**Figure 4 ijms-24-06808-f004:**
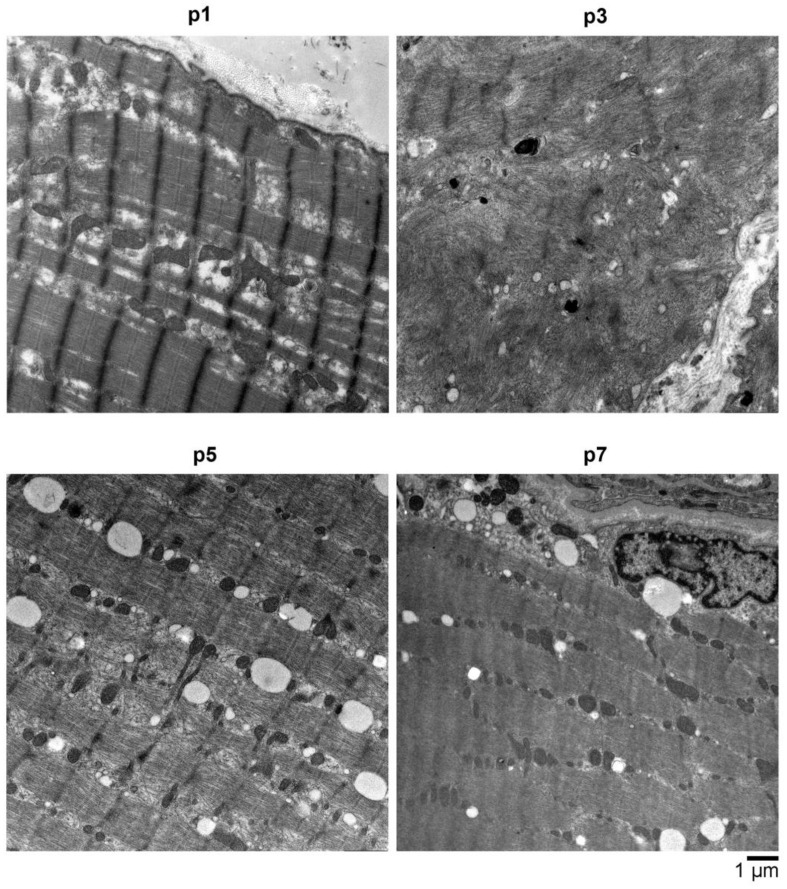
Electron microscopy (EM) analysis of muscle biopsy material obtained from various affected individuals (p1 (p.Ser107Leu), p3 (p.Arg730His), p5 (p.Arg635Leu in *BICD2* and p.Val758Met in *FLNC*), and p7 (p.Lys818Glu in *BICD2* and p.Arg565Gln in *COL6A1*)) showed all had an advanced stage of myofibrillar breakdowns, lesions, and a massive presence of lysosomal vesicles, accompanied by enlarged polymorphic mitochondria and abundant autophagic vacuoles. In most myocytes, the sarcolemma appears with perforations and leaks, and the overall contractile apparatus shows only minor structural integrity. Scale bar: 1 μm.

**Figure 5 ijms-24-06808-f005:**
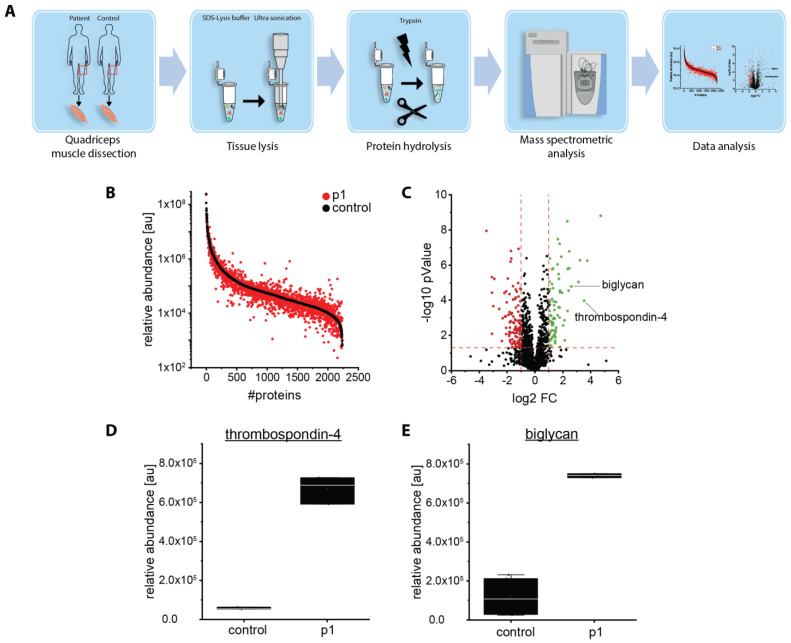
Proteomic studies on a muscle biopsy derived from a *BICD2*-patient (p1): (**A**) Schematic representation of the applied workflow. (**B**) Abundance plot for proteomic profiling data obtained on quadriceps muscle showing the dynamic range of all identified proteins. This is based on their relative quantification of the three highest abundant peptides for each protein, allowing protein comparison within an experiment. All identified proteins of the control (black) are sorted with decreasing abundance while the patient (red) was plotted in the same order to directly compare the different abundances. All identified proteins cover a dynamic range of eight orders of magnitude. (**C**) Volcano plot for proteomic findings obtained in quadriceps muscle highlighting statistically significant increased proteins (green dots) as well as decreased proteins (red dots). FC = fold change. (**D**,**E**) Boxplots of the abundance for thrombospondin-4 and biglycan in muscle. Relative abundance was measured in two healthy controls and p1 in triplicates each. Mean values with standard deviation are displayed.

**Figure 6 ijms-24-06808-f006:**
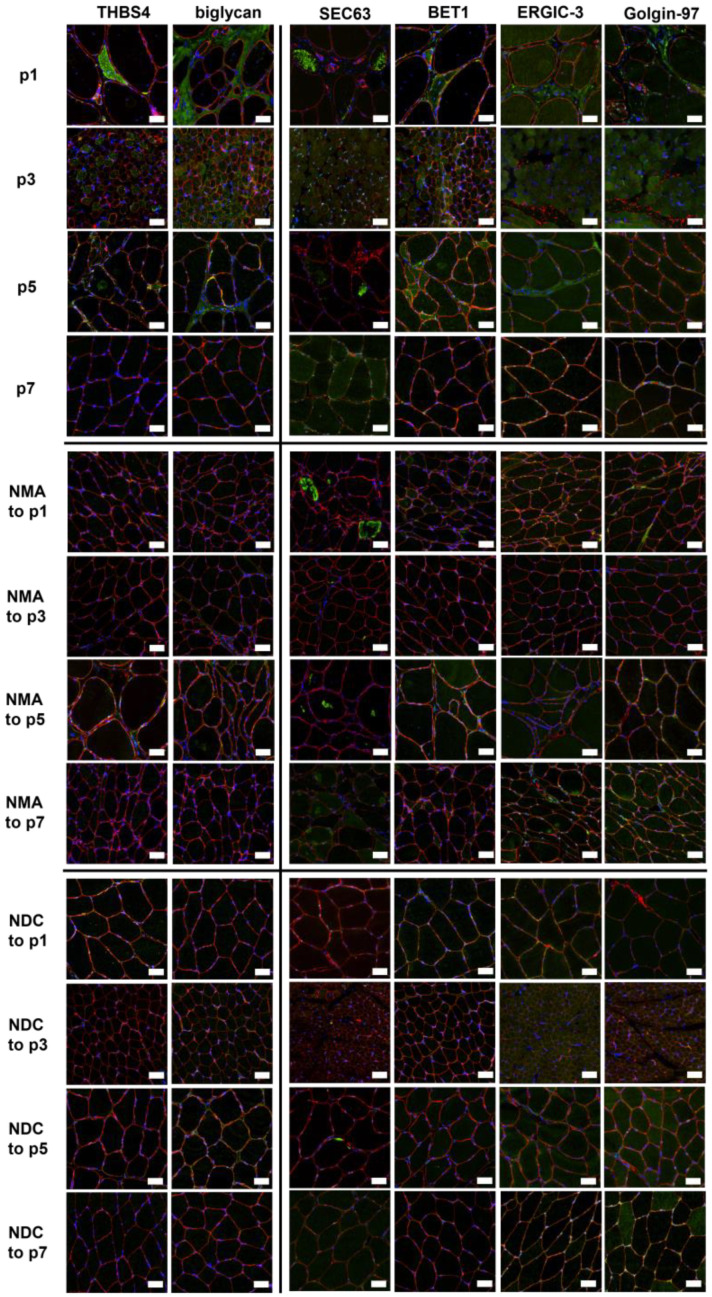
Verification of proteomic data by immunofluorescence analyses. The secretory proteins thrombospondin-4 (THBS4) and biglycan identified by proteomic analysis (**left**) as well as the proteins SEC63, BET1, ERGIC-3, and Golgin-97 associated with ER and (transport to) Golgi (**right**) were stained in p1, p3, p5, and p7 as well as in age-matched patients with neurogenic muscular atrophy (NMA) and normal disease controls (NDC). Co-staining with spectrin (red) was performed to visualize the sarcolemma. Scale bars 50 µm.

**Table 1 ijms-24-06808-t001:** Clinical characteristics of the patients included in this study.

Patient	Mutation	Onset	Pattern of Muscle Weakness	Muscle Atrophy	Creatin-Kinase (CK) Level
p1	Heterozygous c.320C>T; p.(Ser107Leu) Published in [12]	Congenital	Pronounced paresis of distal leg muscles	Calf atrophy, pes cavus	600–650 U/L
p2	Heterozygous c.1195C>T; p.(Arg399Cys)	Early adulthood	Severe paresis pronounced in proximal and distal leg muscles, mild paresis of proximal and distal arm muscles, MRI: adductor muscles spared	Proximal and distal leg muscles	200–300 U/L
p3 (son of p4)	Heterozygous c.2189G>A; p.(Arg730His)	Early childhood	Distal paresis of leg muscles	Distal leg muscles	200–600 U/L
p4 (father of p3)	Heterozygous c.2189G>A; p.(Arg730His)	Early childhood	Distal paresis of leg muscles	Distal leg muscles (right >>left)	600–650 U/L
p5 (mother of p6)	*BICD2*: heterozygous c.1904G>T; p.(Arg635Leu), *FLNC*: heterozygous c.2272G>A; p.(Val758Met)	Early childhood	Pronounced paresis of distal leg muscles, slight proximal paresis of arms and legs	Distal leg muscles and shoulder muscles	150–200 U/L
p6 (daughter of p5)	*BICD2*: heterozygous c.1904G>T; (p.Arg635Leu), *FLNC*: heterozygous c.2272G>A; (p.Val758Met)	Early childhood	Pronounced paresis of distal leg muscles, slight proximal paresis of arms and legs, hyperlordosis	Distal leg muscles	50–100 U/L
p7	*BICD2*: heterozygous c.2452A>G; p.(Lys818Glu), *COL6A1*: heterozygous c.1694G>A; p.(Arg565Gln)	Late adulthood	Proximal paresis of the lower limb muscles	Thigh muscles	200–300 U/L

**Table 2 ijms-24-06808-t002:** Histopathological characteristics of skeletal muscle biopsies obtained during routine diagnostics. IF: immunofluorescence analysis, WB: Western Blot analysis.

Patient	Biopsied Muscle Age at Biopsy	Light Microscopy	Fiber Size	Oxidative Enzyme Reaction	IF/WB Analysis
p1	Vastus lateralis 32 years	Lipofibromatosis, fiber size variation with some angular atrophic fibers, type-1-fiber predominance, grouping of both fiber types, fiber splittings, central nuclei, few necrotic and regenerating fibers, few nuclear clumps	Type 1 fibers: 9–186 µm Type 2 fibers: 4–125 µm	No abnormalities	Reduced: dystrophin 2, dysferlin, alpha-dystroglycan Normal: dystrophin 1 and 3, alpha-sarcoglycan, gamma-sarcoglycan, laminin alpha 2 (80 kD + 300 kD), dysferlin, calpain 2C4
p2	Gastrocnemius, caput mediale 40 years	Mild lipofibromatosis, fiber necrosis and regenerating fibers, central nuclei, vacuolar changes	Type 1 fibers: 16–161 µm Type 2 fibers: 4–136 µm	Few fibers with irregular oxidative enzyme reaction	Aggregates of: desmin myotilin
p3	Quadriceps femoris 19 months	Fiber size variation, few angular atrophic fibers, very few regenerating fibers, very few necrosis	Type 1 fibers: 7.5–17.5 µm Type 2 fibers: 7.5–25 µm	No abnormalities	Reduced: dystrophin 1–3 alpha-dystroglycan Normal: beta-dystroglycan alpha-, beta-, gamma-, and delta-sarcoglycan laminin alpha 2 (80 kD + 300 kD), dysferlin, collagen VI, utrophin, neonatal myosin, calpain 3, dysferlin
p5	Tibialis anterior 55 years	Central nuclei, few fiber splittings, fiber size variation, type1-fiber-predominance	Type 1 fibers: 9–106 µm Type 2 fibers: 19–90 µm	Many fibers with irregular oxidative enzyme reaction	Reduced: dysferlin (normal in WB analysis) Normal dystrophin 2, alpha-sarcoglycan, gamma-sarcoglycan, alpha-dystroglycan, laminin alpha 2 (80 kD + 300 kD), myotilin, caveolin 3, calpain 2C4
p7	Vastus medialis 67 years	Mild lipofibromatosis, slight fiber size variation	Type 1 fibers: 13–116 µm Type 2 fibers: 20–107 µm	No abnormalities	Reduced: dysferlin Normal: Dystrophin 1, 2, and 3, alpha- and gamma-sarcoglycan, alpha-dystroglycan, calpain 2C4

## Data Availability

The proteomic profiling data have been deposited in the ProteomeXchange Consortium via the PRIDE partner repository with the dataset identifier PXD.

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
