# Peer review of "Microscopic and Biochemical Hallmarks of BICD2-Associated Muscle Pathology toward the Evaluation of Novel Variants"

_ijms, 2023, doi:10.3390/ijms24076808_

Round 1
Reviewer 1 Report
A very interesting publication. The authors cover an important topic. The article addresses the link between BICD2 variants and the important problem of neurodegenerative diseases such as spinal muscular atrophy with predominance of the lower extremities (SMALED2) or hereditary spastic paraplegia, muscular atrophy with predominance of the lower extremities (SMALED2) or hereditary spastic paraplegia. I believe the topic is extremely important as it relates to neurodegenerative diseases, which are an important medical problem. The article highlights a specific BICD2 variant, which may help in further research and, consequently, in the treatment of these conditions. The methodology doesn't need to improve and the conclusions are in line with the work. The tables are correct. Figures and photographs of preparations deserve attention and praise. Accurate references should be expanded, as the bibliography is too small.
The only comments I have are:
-the discussion is too short
-poor bibliography - it could be expanded.
After making such changes, the publication is suitable for opubilization in IJMS.
Author Response
A very interesting publication. The authors cover an important topic. The article addresses the link between BICD2 variants and the important problem of neurodegenerative diseases such as spinal muscular atrophy with predominance of the lower extremities (SMALED2) or hereditary spastic paraplegia, muscular atrophy with predominance of the lower extremities (SMALED2) or hereditary spastic paraplegia. I believe the topic is extremely important as it relates to neurodegenerative diseases, which are an important medical problem. The article highlights a specific BICD2 variant, which may help in further research and, consequently, in the treatment of these conditions. The methodology doesn't need to improve and the conclusions are in line with the work. The tables are correct. Figures and photographs of preparations deserve attention and praise. Accurate references should be expanded, as the bibliography is too small.
Point of concern 1: the discussion is too short.
Reply 1: We agree that more aspects of BICD2-associated disorders should be discussed, including underlying pathomechanisms and genetic landscapes. As different clinical phenotypes have been described in the context of BICD2-mutations, we added a section focussing to the pathomechanisms leading to SMALED2 and HSP to the revised version of the discussion.
Point of concern 2: poor bibliography - it could be expanded.
Reply 2: In the context of the new aspects added in the discussion section, we also complemented the bibliography by adding corresponding references.

Reviewer 2 Report
Dear Authors and Editors
Thank you for the invitation to review this original research. This study identifies the biglycan and thrombospondin-4 may serve as tissue pathogenicity markers and might be linked to perturbed protein secretion based on impaired vesicular transportation. The manuscript is clear, relevant, and presented in a well-structured manner. Furthermore, it is scientifically sound and has an appropriate design. The results are reproducible depending on the details given in the methods. The tables and images are appropriate, correctly display the data, and are easy to interpret and understand. Data are interpreted appropriately and consistently throughout the manuscript. The conclusions are consistent with the evidence and arguments presented.
Reviewer 3 Report
The article by Unger Andreas et al. provides an extensive and comprehensive characterization of effects of BICD2 changes in neurodegenerative disorders. The work is interesting, but I just have a few small concerns:
- In the 2.2 Genetic analysis revealed a known pathgenic ans six novel sequence variants in BICD2
The authors report that they have used prediction programs for annotation of the variants. Could they specify which ones? SIFT, Polyphen, Mutation taster, PROVEAN? Were the ACMG criteria used?
- In Table 2 of the manuscript the authors reported that the muscle biopsy was performed at age 42 for patient 1. In the article (reference 12) which they cited as the paper where this mutation was published they mentioned the age of 32. Is there a mistake? What age was the patient? Also of patients 3 and 4 of family 3 is no MRI available?
- In the 4.3 Genetic analyses
I would suggest the authors to report the filtering algorithm used. In addition, was the segregation analysis not possible for all families?
Author Response
The article by Unger Andreas et al. provides an extensive and comprehensive characterization of effects of BICD2 changes in neurodegenerative disorders. The work is interesting, but I just have a few small concerns:
Point of concern 1: In the 2.2 “Genetic analysis” revealed a known pathogenic and six novel sequence variants in BICD2.
The authors report that they have used prediction programs for annotation of the variants. Could they specify which ones? SIFT, Polyphen, Mutation taster, PROVEAN? Were the ACMG criteria used?
Reply 1: For annotation of the variants described, we used several the prediction programs (MutationTaster, PROVEAN, SIFT). A table with additional information including prediction programs, ACMG criteria and ClinVar database entries was added as supplementary Table S1 to the revised version of the manuscript.
Point of concern 2: In Table 2 of the manuscript the authors reported that the muscle biopsy was performed at age 42 for patient 1. In the article (reference 12) which they cited as the paper where this mutation was published they mentioned the age of 32. Is there a mistake? What age was the patient? Also, of patients 3 and 4 of family 3 is no MRI available?
Reply 2: We apologize for this mistake. The muscle biopsy in the index patient in family 1 was performed at the age of 32, not 42. We corrected this point in the revised manuscript (Table 2, page 5). An MRI-scan of patients 3 and 4 of family 3 would have been much appreciated. Unfortunately, the family has not been available for further examinations due to personal reasons other than the ones for which the results are presented in this study.
Point of concern 3: In the 4.3 Genetic analyses - I would suggest the authors to report the filtering algorithm used. In addition, was the segregation analysis not possible for all families?
Reply 3: For genetic analyses performed for routine diagnostics, genetic variants with an allele frequency of more than one percent were discharged, as long as they were not annotated as pathogenic or likely pathogenic in HGMD or ClinVar.
Segregation analyses were performed for all families except for family 2. The sentence was rephrased accordingly in section 4.3, page 14.
For better understanding, the genetics part of the manuscript (esp. the sections 2.2. and 4.3) was revised and rephrased, and the supplementary Table S1 was added in order to provide additional information regarding our genetic analyses of the variants presented in this study.

Reviewer 4 Report
BICD2- variants have been associated with neurodegenerative disorders. BICD2 mutations have been reported to be implicated in myopathy. The authors approach provides new insights into pathophysiology of BICD2-associated disorders confirming a primary muscle cell vulnerability. Biglycan and thrombospondin-4 have been identified as tissue pathogenicity markers.
Major Comments
1. Please discuss regarding the patho-mechanism of how mutations in BICD2 can cause SMALED2 and rarely HSP-like phenotype.
2. Please include morphological analysis of semi-thin ventral root sections for seeing acute axon degeneration profiles, if possible.
3. THBS4 stimulation activates β-catenin 1 pathway. Hence, it will be a good idea to check the levels of β-catenin in your study.
Minor Comments
1. Please provide a summary or graphical abstract of the entire research study for easy understanding.
Author Response
BICD2- variants have been associated with neurodegenerative disorders. BICD2 mutations have been reported to be implicated in myopathy. The authors approach provides new insights into pathophysiology of BICD2-associated disorders confirming a primary muscle cell vulnerability. Biglycan and thrombospondin-4 have been identified as tissue pathogenicity markers.
Point of concern 1: Please discuss regarding the pathomechanism of how mutations in BICD2 can cause SMALED2 and rarely HSP-like phenotype.
Reply 1: We thank the reviewer for this important comment and agree that it is important to note that different clinical phenotypes have been described in the context of pathogenic BICD2 variants and are associated underlying pathomechanisms. This important aspect has been added to the discussion section of the revised version of the manuscript.
Point of concern 2: Please include morphological analysis of semi-thin ventral root sections for seeing acute axon degeneration profiles, if possible.
Reply 2: This is an important point, and it would indeed be interesting to see if there are differences in axon degeneration between patients with a muscular and those with a SMALED2- or HSP phenotype. Due to the fact that we do not have post-mortem material of BICD2-patients, this point of concern cannot be addressed in the framework of the study presented but might represent an important issue for further studies for instance such focussing on neuropathological changes in a Bicd2-/- mouse model.
Point of concern 3: THBS4 stimulation activates β-catenin 1 pathway. Hence, it will be a good idea to check the levels of β-catenin in your study.
Reply 3: The influence of the observed THBS4-pathology on the beta-catenin 1 pathway would indeed be interesting to investigate (including its influence on the cytoskeleton) to further elucidate the myopathology associated with pathogenic BICD2 variants. Given that the timeframe for the resubmission of our manuscript is too short to perform additional immunofluorescence studies in this regard (required antibodies are not available in our laboratory), we were carefully filtering our proteomic data for proteins involved in the beta-catenin 1 pathway. This approach revealed that in the muscle tissue of our patient, beta-catenin itself as well as proteins involved in the Wnt/beta-catenin pathway were not significantly altered. Nevertheless, this is an intriguing aspect, which should be addressed in proteomic and immunofluorescence analysis of additional patients with BICD2-associated disorders.
Point of concern 4: Please provide a summary or graphical abstract of the entire research study for easy understanding.
Reply 4: We thank the reviewer for this important suggestion and - for better understanding - added a graphical abstract of the research performed in the study presented (Figure 1, page 2).
